# Focus on the Common Good: Group Distributional Robustness Follows

Vihari Piratla[*], Praneeth Netrapalli[2], and Sunita Sarawagi[1]

[1]Indian Institute of Technology, Bombay
[2]Google Research, India

## Abstract

We consider the problem of training a classification model with group annotated training data. Recent work has established that, if there is distribution shift across different groups, models trained using the standard empirical risk minimization (ERM) objective suffer from poor performance on minority groups and that group distributionally robust optimization (Group-DRO) objective is a better alternative. The starting point of this paper is the observation that though Group-DRO performs better than ERM on minority groups for *some* benchmark datasets, there are *several other* datasets where it performs much worse than ERM. Inspired by ideas from the closely related problem of domain generalization, this paper proposes a new and simple algorithm that explicitly encourages learning of features that are shared across various groups. The key insight behind our proposed algorithm is that while Group-DRO focuses on groups with worst regularized loss, focusing instead, on groups that enable better performance even on other groups, could lead to learning of shared/common features, thereby enhancing minority performance beyond what is achieved by Group-DRO. Empirically, we show that our proposed algorithm matches or achieves better performance compared to strong contemporary baselines including ERM and Group-DRO on standard benchmarks on both minority groups and across all groups. Theoretically, we show that the proposed algorithm is a descent method and finds first order stationary points of smooth nonconvex functions. Our code and datasets can be found at this URL.

## 1 Introduction

It is by now well known (Sagawa et al., 2020a) that, in several settings with subpopulation shift, large neural networks trained using empirical risk minimization (ERM) objective heavily sacrifice prediction accuracy on small groups/subpopulations of the data, in order to achieve a high overall prediction accuracy. One of the key reasons for this behavior of ERM is the existence of spurious correlations between labels and some features on majority groups which are either nonexistent, or worse oppositely correlated, on the minority groups (Sagawa et al., 2020a). In such cases, ERM exploits these spurious correlations to achieve high accuracy on the majority groups, thereby suffering from poor performance on minority groups. Consequently, this is a form of unfairness, where accuracy on minority subpopulation or group is being heavily sacrificed to achieve near perfect accuracy on majority groups.

More concretely, we are given training examples, stratified non-uniformly in to multiple subpopulations, also called groups. Our goal is to train a model that generalizes to *all training groups without severely sacrificing accuracy on the minority groups*. This problem, which is also known as the sub-population shift problem, was popularized by Sagawa et al. (2020a), and is well-studied in the literature (Sagawa et al., 2020a;b; Menon et al., 2021; Zhai et al., 2021; Goel et al., 2020; Bao et al., 2021). Among all algorithms proposed for this problem, one of the most influential ones is the Group Distributionally Robust Optimization (Group-DRO) method (Sagawa et al., 2020a), which at every update step focuses on the group with the highest regularized loss. While Group-DRO has

---

[*]viharipiratla@gmail.com

been shown to obtain reduction in worst-group error over ERM on *some* benchmark datasets, it has obvious but practically relevant failure modes such as when different groups have different amounts of label noise. In such a case, Group-DRO ends up focusing on the group(s) with the highest amount of label noise, thereby obtaining worse performance on all groups. In fact, on *several other* datasets with subpopulation shift, Group-DRO performs poorly compared to ERM (Koh et al., 2020).

The key issue with Group-DRO's strategy of just focusing on the group with the highest training loss is that, it is uninformed of the inter-group interactions. Consequently, the parameter update using the group with the highest training loss may increase the loss on other groups. In fact, this observation also reveals that Group-DRO is not properly addressing the issue of spurious correlations, which have been identified by prior work as a key reason behind the poor performance of ERM (Sagawa et al., 2020a). Inspired by ideas from the closely related problem of *domain generalization* (Sun & Saenko, 2016; Shankar et al., 2018; Arjovsky et al., 2019), we hypothesize that modeling inter-group interactions is essential for properly addressing spurious correlations and sub-population shift problem. More concretely, in each update step, we propose the following simple strategy to decide which group to train on:

*Train on that group whose gradient leads to largest decrease in average training loss over all groups.*

In other words, our strategy focuses on the group that contributes to the common good. We call the resulting algorithm *Common Gradient Descent* (CGD). We show that CGD is a sound optimization algorithm as it monotonically decreases the macro/group-average loss, and consequently finds first order stationary points. CGD's emphasis on common gradients that lead to improvements across all groups, makes it robust to the presence of spurious features, enabling it to achieve better generalization across all groups compared to usual gradient descent. We present insights on why CGD may perform better than Group-DRO by studying simple synthetic settings. Subsequently, through empirical evaluation on seven real-world datasets–which include two text and five image tasks with a mix of sub-population and domain shifts, we demonstrate that CGD either matches or obtains improved performance over several existing algorithms for robustness that include: ERM, PGI (Ahmed et al., 2021), IRM (Arjovsky et al., 2019), and Group-DRO.

## 2 METHOD

**Problem statement**    Let $\mathcal{X}$ and $\mathcal{Y}$ denote the input and label spaces respectively. We assume that the training data comprises of $k$ groups from a set $\mathcal{G}$ where each $i \in \mathcal{G}$ include $n_i$ instances from a probability distribution $P_i(\mathcal{X}, \mathcal{Y})$. In addition to the label $y_j \in \mathcal{Y}$, each training example $x_j \in \mathcal{X}$ is also annotated by the group/subpopulation $i \in \mathcal{G}$ from which it comes. The group annotations are available only during training and not during testing time, hence the learned model is required to be group-agnostic, i.e., it may not use the group label at test time. The number of examples could vary arbitrarily between different groups in the train data. We refer to groups $\{i\}$ with (relatively) large $n_i$ as majority groups and those with small $n_i$ as minority groups. We use the terms group and sub-population interchangeably. Our goal is to learn a model that performs well on all the groups in $\mathcal{G}$. Following prior work (Sagawa et al., 2020a), we use two metrics to quantify performance: *worst group accuracy* denoting the minimum test accuracy across all groups $i \in \mathcal{G}$ and *average accuracy* denoting the average test accuracy across *all examples belonging to all groups*.

We denote with $\ell_i$, the average loss over examples of group $i$ using a predictor $f_\theta$, $\ell_i(\theta) = \mathbb{E}_{(x,y) \sim P_i(\mathcal{X}, \mathcal{Y})} \mathcal{L}(x, y; f_\theta)$, for an appropriate classification loss $\mathcal{L}$, a parameterized predictor $f_\theta$ with parameter $\theta$. We refer to the parameters at step 't' of training by $\theta^t$.

The ERM training algorithm learns parameters over the joint risk, which is the population weighted sum of the losses: $\sum_i n_i \ell_i / \sum_i n_i$. However, as described in Section 1 when there are spurious correlations of certain features in the majority groups, ERM's loss cannot avoid learning of these spurious features. This leads to poor performance on minority groups if these features are either nonexistent or exhibit opposite behavior on the minority groups. A simple alternative, called ERM-UW, is to reshape the risk function so that the minority groups are up-weighted, for some predefined group-specific weights $\alpha$. ERM-UW while simple, however, could overfit to the up-weighted minority groups. Sagawa et al. (2020a) studied an existing robust optimization algorithm (Duchi & Namkoong, 2018) in the context of deep learning models. They proposed a risk reshaping method called Group-DRO, which at any update step trains on the group with the highest loss, as shown

below.

$$\text{Group-DRO update step:} \quad j^* = \arg\max_j \ell_j(\theta) \quad \text{and} \quad \theta^{t+1} = \theta^t - \eta \nabla_\theta \ell_{j^*}.$$

Training on only high loss groups avoids overfitting on the minority group since we avoid zero training loss on any group while the average training loss is non-zero. However, this approach of focusing on the worst group is subject to failure when groups have heterogeneous levels of noise and transfer. In Section 4, we will illustrate these failure modes.

Instead, we consider an alternate strategy for picking the training group: the group when trained on, most minimizes the overall loss across *all* groups. Let $g_i = \nabla_\theta \ell_i$ represent the gradient of the group $i$, we pick the desired group as:

$$j^* = \arg\min_j \sum_i \ell_i(\theta^t - \eta \nabla_\theta \ell_j(\theta^t)) \qquad \approx \arg\max_j \sum_i g_i^T g_j \quad \text{[First-order Taylor]} \quad (1)$$

The choice of the training group based on gradient inner-product can be noisy. To counteract, we smooth the choice of group with a weight vector $\alpha^t \in \Delta^{k-1}$, at step t, and also regularize between consecutive time steps. The amount of regularization is controlled by a hyperparameter: $\eta_\alpha$. Our parameter update at step $t+1$ takes the following form.

$$\alpha^{t+1} = \arg\max_{\alpha \in \Delta^{k-1}} \sum_i \alpha_i \langle g_i, \sum_j g_j \rangle - \frac{1}{\eta_\alpha} KL(\alpha, \alpha^t) \quad (2)$$

$$\theta^{t+1} = \theta^t - \eta \sum_j \alpha_j^{t+1} g_j(\theta_t).$$

The update of $\alpha$ in equation 2 can be solved in closed form using KKT first order optimality conditions and rearranging to get:

$$\alpha_i^{t+1} = \frac{\alpha_i^t \cdot \exp\left(\eta_\alpha \langle g_i, \sum_j g_j \rangle\right)}{\sum_s \alpha_s^t \cdot \exp\left(\eta_\alpha \langle g_s, \sum_j g_j \rangle\right)}. \quad (3)$$

Pseudocode for the overall algorithm is shown in Algorithm 1.

---

**Algorithm 1** CGD Algorithm

1: **Input:** Number of groups: $k$, Training data: $\{(x_j, y_j, i) : i \in [k], j \in [n_i]\}$, Step sizes: $\eta_\alpha, \eta$
2: Initialize $\theta^0, \alpha^0 = \left(\frac{1}{k}, \cdots, \frac{1}{k}\right)$
3: **for** $t = 1, 2, \cdots,$ **do**
4:     **for** $i \in \{1, \cdots, k\}$ **do**
5:         $\alpha_i^{t+1} \leftarrow \alpha_i^t \exp(\eta_\alpha \nabla \ell_i(\theta^t)^\top \sum_{s \in [k]} \nabla \ell_s(\theta^t))$
6:     **end for**
7:     $\alpha_i^{t+1} \leftarrow \alpha_i^{t+1}/\|\alpha^{t+1}\|_1 \quad \forall i \in [1 \ldots k]$            ▷ Normalize
8:     $\theta^{t+1} \leftarrow \theta^t - \eta \sum_{i \in \{1, \cdots, k\}} \alpha_i^{t+1} \nabla \ell_i(\theta^t)$         ▷ Update parameters
9: **end for**

---

The scale of the gradients can vary widely depending on the task, architecture and the loss landscape. As a result, the $\alpha$ update through equation 3 can be unstable and tuning of the $\eta_\alpha$ hyperparam tedious. Since we are only interested in capturing if a group transfers positively or negatively to others, we retain the cosine-similarity of the gradient dot-product but control their scale through $\ell(\theta)^p$ for some $p > 0$. That is, we set the gradient: $\nabla \ell_i(\theta^t)$ to $\frac{\nabla \ell_i(\theta^t)}{\|\nabla \ell_i(\theta^t)\|} \ell_i(\theta)^p$. In our implementation, we use $p = 1/2$. In Appendix A, we discuss in more detail the range of gradient norm, why we scale the gradient by loss, and how we pick the value of p. Finally, the $\alpha^{t+1}$ update of equation 3 is replaced with equation 4. When we assume the groups do not interact, i.e. $\langle g_i, g_j \rangle = 0 \quad \forall i \neq j$, then our update (equation 4) matches Group-DRO.

$$\alpha_i^{t+1} = \frac{\alpha_i^t \exp(\eta_\alpha \sum_j \sqrt{\ell_i \ell_j} \cos(g_i, g_j))}{\sum_s \alpha_s^t \exp(\eta_\alpha \sum_j \sqrt{\ell_s \ell_j} \cos(g_s, g_j))}. \quad (4)$$

**Group Adjustment** The empirical train loss underestimates the true loss by an amount that is inversely proportional to the population size. Owing to the large group population differences, we expect varying generalization gap per group. Sagawa et al. (2020a) adjusts the loss value of the groups to correct for these generalization gap differences as $\ell_i = \ell_i + C/\sqrt{n_i}, \quad \forall i.$, where $C > 0$ is a hyper-parameter. We apply similar group adjustments for CGD as well. The corrected loss values simply replace the $\ell_i$ of Line 5 in Algorithm 1.

## 3 CONVERGENCE ANALYSIS

In this section, we will show that CGD is a sound optimization algorithm by proving that it monotonically decreases the function value and finds first order stationary points (FOSP) for bounded, Lipschitz and smooth loss functions. We now define the notion of $\epsilon$-FOSP which is the most common notion of optimality for general smooth nonconvex functions.

**Definition 1.** *A point $\theta$ is said to be an $\epsilon$-FOSP of a differentiable function $f(\cdot)$ if $\|\nabla f(\theta)\| \leq \epsilon$.*

In the context of our paper, we consider the following cumulative loss function: $\mathcal{R}(\theta) := \frac{1}{k}\sum_i \ell_i(\theta)$. This is called the macro/group-average loss. We are now ready to state the convergence guarantee for our algorithm.

**Theorem 1.** *Suppose that (i) each $\ell_i(\cdot)$ is $G$-Lipschitz, (ii) $\mathcal{R}(\cdot)$ is $L$-smooth and (iii) $\mathcal{R}(\cdot)$ is bounded between $-B$ and $B$. Suppose further that Algorithm 1 is run with $\eta = 2\sqrt{\frac{B}{LG^2T}}$ and $\eta_\alpha = \sqrt{\frac{BL}{G^6T}}$. Then, Algorithm 1 will find an $\epsilon$-FOSP of $\mathcal{R}(\theta)$ in $O\left(\frac{BLG^2}{\epsilon^4}\right)$ iterations.*

In other words, the above result shows that Algorithm 1 finds an $\epsilon$-FOSP of the cumulative loss in $O\left(\epsilon^{-4}\right)$ iterations. We present a high level outline of the proof here and present the complete proof of Theorem 1 in Appendix B.

*Proof outline of Theorem 1.* Considering the $t^{\text{th}}$ iteration where the iterate is $\theta^t$ and mixing weights are $\alpha^t$. Using the notation in Section 2, let us denote $\mathcal{R}(\theta) = \frac{1}{k}\sum_i \ell_i(\theta)$, $g_i = \nabla \ell_i(\theta^t)$ and $g = \frac{1}{k}\sum_i g_i$. The update of our algorithm is given by:

$$\alpha_i^{t+1} = \frac{\alpha_i^t \cdot \exp\left(\eta_\alpha \langle g_i, g \rangle\right)}{Z} \tag{5}$$

$$\theta^{t+1} = \theta^t - \eta \sum_i \alpha_i^{t+1} g_i, \tag{6}$$

where $Z = \sum_j \alpha_j^t \cdot \exp\left(\eta_\alpha \langle g_j, g \rangle\right)$. Let us fix $\alpha^* = (1/k, \cdots, 1/k) \in \mathbb{R}^k$ and use $KL(p,q) = \sum_i p_i \log \frac{p_i}{q_i}$ to denote the KL-divergence between $p$ and $q$. Noting that the update equation 5 on $\alpha$ corresponds to mirror descent steps on the function $\langle g, \sum_i \alpha_i g_i \rangle$ and using mirror descent analysis, we obtain:

$$KL(\alpha^*, \alpha^{t+1}) \leq KL(\alpha^*, \alpha^t) + \left(\sum_i \alpha_i^t \eta_\alpha \langle g_i, g \rangle\right) + \left(\eta_\alpha G^2\right)^2 - \eta_\alpha \|g\|^2$$

$$\Rightarrow -\sum_i \alpha_i^t \langle g_i, g \rangle \leq -\|g\|^2 + \frac{KL(\alpha^*, \alpha^t) - KL(\alpha^*, \alpha^{t+1})}{\eta_\alpha} + \eta_\alpha G^4.$$

Using monotonicity of the $\exp$ function, we further show that $\sum_i \alpha_i^t \langle g_i, g \rangle \leq \sum_i \alpha_i^{t+1} \langle g_i, g \rangle$.

Using smoothness of $\mathcal{R}$ and update equation 6, we then show:

$$\mathcal{R}(\theta^{t+1}) \leq \mathcal{R}(\theta^t) + \eta\left(-\|g\|^2 + \frac{KL(\alpha^*, \alpha^t) - KL(\alpha^*, \alpha^{t+1})}{\eta_\alpha} + \eta_\alpha G^4\right) + \frac{\eta^2 LG^2}{2}$$

$$\Rightarrow \|g\|^2 \leq \frac{\mathcal{R}(\theta^t) - \mathcal{R}(\theta^{t+1})}{\eta} + \frac{KL(\alpha^*, \alpha^t) - KL(\alpha^*, \alpha^{t+1})}{\eta_\alpha} + \eta_\alpha G^4 + \frac{\eta LG^2}{2}.$$

Summing the above inequality over timesteps $t = 1, \cdots, T$, and using the parameter choices for $\eta$ and $\eta_\alpha$ proves the theorem. $\square$

## 4 QUALITATIVE ANALYSIS

In this section, we look at simple multi-group training scenarios to derive insights on the difference between Group-DRO and CGD. We start with a simple setup where one of the groups is noisy in Section 4.1. In Section 4.2, we look at a setting with uneven inter-group transfer. In Section 4.3, we study a spurious correlation setup. For all the settings, we train a linear binary classifier model for 400 epochs with batch SGD optimizer and learning rate 0.1. Unless otherwise stated, the number of input dimensions is two and the two features $x_1, x_2$ are sampled from a standard normal, with the binary label $y = \mathbb{I}[x_1 + x_2 > 0]$, and number of groups is three. For each setting we will inspect the training weight ($\alpha_i^t$) assigned by each algorithm per group $i$ at epoch $t$ (Figure 1, 3, 5). The plots are then used to draw qualitative judgements. See Appendix C for descriptive plots.

We also present parallel settings with MNIST images and deep models (ResNet-18) to stress that our observations are more broadly applicable. Appendix D presents the details of this setup.

### 4.1 LABEL NOISE SETUP: *Group-DRO focuses on noisy groups.*

We induced label noise in the first group, only during training, by flipping labels for randomly picked 20% examples. The first and second group formed the majority with a population of 450 examples each and the minority last group had 100 examples. The noisy first group and the two subsequent groups are referred to as Noisy-Majority, Clean-Majority, and Clean-Minority. Due to noise in the first group, and since we cannot overfit in this setup, the loss on the first group was consistently higher than the other two. As a result, Group-DRO trained only on the first group (Fig 1a), while CGD avoided overly training on the noisy first group (Fig 1b). This results in not only lower worst-case error

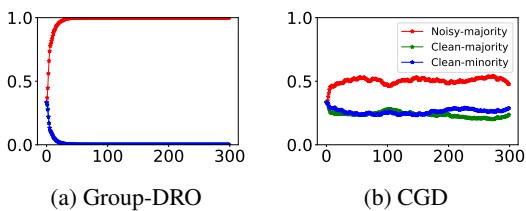

(a) Group-DRO          (b) CGD

Figure 1: Noise-Simple. Training weight ($\alpha$) of groups vs number of epochs.

but more robust training (Noise-Simple column, Table 1). The variance of the learned parameters across six runs drops from 1.88 to 0.32 using CGD.

Table 2 (Noise-MNIST column) shows the same trend under similar setup on the MNIST dataset with the ResNet-18 model. These experiments demonstrate that Group-DRO's method of focusing on the group with the worst training loss is sub-optimal when groups have heterogeneous label noise or training difficulty. CGD's weights that are informed by inter-group interaction provide greater in-built robustness to group-specific noise.

### 4.2 UNEVEN INTER-GROUP SIMILARITY: *CGD focuses on central groups*

Here we simulated a setting such that the optimal classifier for the first, third group is closest to the second. The label (y) was set to a group-specific deterministic function of the input $(x_1, x_2)$; for the first group the mapping function was $y = \mathbb{I}[x_1 > 0]$ (which was rotated by 30 degrees for the two subsequent groups), for the second group it was $y = \mathbb{I}[0.87x_1 + 0.5x_2 > 0]$, and for the third group it was $y = \mathbb{I}[0.5x_1 + 0.87x_2 > 0]$. The angle between the classifiers is a proxy for the distance between the groups. We refer to the groups in the order as *Left, Center, Right*, and their respective training population was: 499, 499, 2.

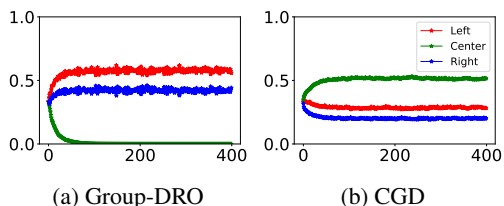

(a) Group-DRO          (b) CGD

Figure 3: Rot-Simple. Training weight ($\alpha$) of groups vs number of epochs.

The optimal classifier that generalizes equally well to all the groups is the center classifier. Group-DRO, CGD, differed in how they arrive at the center classifier: Group-DRO assigned all the training mass equally to the Left and Right groups (Figure 2a), while CGD trained, unsurprisingly, mostly

on the Center group that has the highest transfer (Figure 2b). Group-DRO up-weighted the noisy minority group (2 examples) and is inferior when compared with CGD's strategy, which is reflected by the superior performance on the test set shown in Rotation-Simple column of Table 1. On the MNIST dataset we repeated a similar experiment by rotating digits by 0, 30, and 60 degrees, and found CGD to provide significant gains over Group-DRO (Rotation-MNIST column of Table 2). These experiments demonstrate the benefit of focusing on the central group even for maximizing worst case accuracy.

### 4.3 SPURIOUS CORRELATIONS SETUP: *CGD focuses on groups without spurious correlations*

Here we create spurious correlations by adding a third feature that takes different values across the three different groups whose sizes are 490, 490, and 20 respectively. The third feature was set to the value of the label y on the first group, 1-y on the third group, and set to y or 1-y w.p. 0.6 on the second group. We perturbed the first two features of the first group examples such that they predict label correctly only for 60% of the examples. Therefore, the net label correlation of the first two and the third feature is about the same: 0.8. We refer to the first and

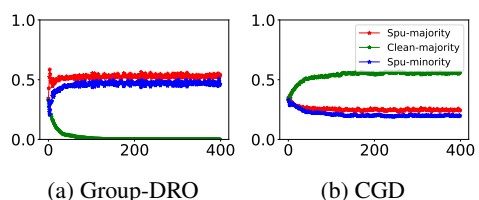

(a) Group-DRO      (b) CGD

Figure 5: Spurious-Simple. Training weight ($\alpha$) of groups vs number of epochs.

last groups as Spurious-majority, Spurious-minority–since they both contain an active spurious third feature–and the second as the clean majority.

Group-DRO balances training between the two spurious groups (Figure 4a), while CGD focuses mostly on the clean group (Figure 4b). Both the training strategies could avoid the spurious third feature, however, the training strategy of Group-DRO is sub-optimal since we up-weight the small, hence noisy, spurious minority group. The better test loss of CGD in Table 1 (Spurious-Simple column) further verifies the drawback of Group-DRO's training strategy. On the MNIST dataset too we observed similar trends by creating spurious correlation using colors of the digit (Spurious-MNIST column of Table 2).

More generally, whenever a relatively clean group with low spurious correlation strength is present, and when the train groups represent both positive and negative spurious correlations, we expect CGD to focus on the group with the least spurious correlation since it is likely to have greater inter-group similarity.

| Task → | Noise-Simple | | Rotation-Simple | | Spurious-Simple | |
|---|---|---|---|---|---|---|
| Alg↓ | Variance | Worst Loss | Variance | Worst Loss | Variance | Worst Loss |
| Group-DRO | 1.88 | 0.35 (0.03) | 0.41 | 0.77 (0.14) | 0.17 | 0.70 (0.16) |
| CGD | 0.32 | 0.25 (0.02) | 0.08 | 0.59 (0.05) | 0.04 | 0.43 (0.06) |

Table 1: Comparison of Group-DRO, CGD across the three simple settings. Worst loss is the worst group binary cross entropy loss on the test set, and averaged over six seeds, shown in parenthesis is the standard deviation. Variance column shows the variance of $L_\infty$ normalized solution across the six runs. CGD has lower variance and test loss when compared with Group-DRO.

| Alg. ↓ | Noise-MNIST | Rotation-MNIST | Spurious-MNIST |
|---|---|---|---|
| Group-DRO | 86.0 (1.0), 85.9 (1.0) | 90.5 (0.2), 80.6 (1.1) | 95.4 (0.2), 95.0 (0.4) |
| CGD | 88.9 (1.0), 88.8 (1.0) | 92.1 (0.6), 85.0 (0.7) | 96.7 (0.1), 96.4 (0.3) |

Table 2: Average, worst group accuracy on the test split. Shown in parenthesis is the standard deviation. All numbers are aggregated over three runs. More details of the setup in Appendix D

## 5 EXPERIMENTS

We compare CGD with state of the art methods from group-distribution shift and domain generalization literature. We enumerate datasets in Section 5.1, discuss the baselines and present implementation details in Section 5.2, and finally present our results in Section 5.3.

### 5.1 DATASETS

We evaluated on eight datasets, which include two synthetic datasets with induced spurious correlations: ColoredMNIST, WaterBirds; two real-world datasets with known spurious correlations: CelebA, MultiNLI; four WILDS (Koh et al., 2020) datasets with a mix of sub-population and domain shift. Table 3 summarises all the eight datasets. We describe all the datasets in Appendix E. For more details on WaterBirds, CelebA, MultiNLI, we point the reader to the original paper (Sagawa et al., 2020a), and Koh et al. (2020) for details on the WILDS datasets. We discuss how some of the real-world datasets relate to our synthetic setting in Appendix H.

| Dataset | # labels | # groups | Population type | Type | Size | Worst ratio |
|---------|----------|----------|----------------|------|------|-------------|
| Colored MNIST | 2 | 2 | Label×Group | Image | 50K | 1000 |
| WaterBirds | 2 | 2 | Label×Group | Image | 4.8K | 62.5 |
| CelebA | 2 | 2 | Label×Group | Image | 162K | 51.6 |
| MultiNLI | 3 | 2 | Label×Group | Text | 200K | 44.3 |
| CivilComments-WILDS | 2 | 2 | Label×Group | Text | 270K | 74.5 |
| PovertyMap-WILDS | real | 13 | Group | Image | 10K | 5.9 |
| FMoW-WILDS | 62 | 11 | Group | Image | 80K | 16.0 |
| Camelyon17-WILDS | 2 | 3 | Group | Image | 3K | 2.5 |

Table 3: Summary of the datasets we used for evaluation. The Size column shows the number of training instances. The Worst ratio column shows the ratio of the size of the largest train group to the smallest; worst ratio reflects loosely the strength of the spurious correlations.

### 5.2 EXPERIMENT DETAILS

**Baselines**
*ERM*: Simple descent on the joint risk.
*ERM-UW*, *Group-DRO*: risk reshaping baselines described in Section 2. With ERM-UW, instead of simply up-weighting all the groups equally, we set the group weighting parameter $\alpha_i$ of a group $i$ as a function of its size. For some non-negative constant C, $\alpha_i \propto \exp(C/\sqrt{n_i})$.
*PGI* (Ahmed et al., 2021), predictive group invariance, penalizes the divergence between predictive probability distributions among the training groups, and was shown to be effective for in-domain as well as out-domain generalization. We provide more details about the algorithm in Appendix F.

On the WILDS dataset, we also compare with two algorithms from Domain Generalization literature: *CORAL* (Sun & Saenko, 2016), *IRM* (Arjovsky et al., 2019). Domain Generalization methods are relevant since they could generalize to any domain including the seen train domains.

**Implementation Details and Evaluation Metric:** We use the codestack[1] released with the WILDS (Koh et al., 2020) dataset as our base implementation. Our code can be found at this URL. We report the average and standard deviation of the evaluation metric from at least three runs. If not otherwise stated explicitly, we report worst accuracy over the groups evaluated on the test split.

**Hyperparameters:** We search $C$: the group adjustment parameter for Group-DRO, CGD, and group weighting parameter for ERM-UW, over the range $[0, 20]$. We tune $C$ only for WaterBirds, CelebA and MultiNLI datasets for a fair comparison with Group-DRO. For other dataset, we set C to 0 because group adjustment tuning did not yield large improvements on many datasets. The step size parameter of Group-DRO, CGD, and $\lambda$ penalty control parameter of PGI, is picked from $\{1, 0.1, 1e\text{-}2, 1e\text{-}3\}$. We follow the same learning procedure of Sagawa et al. (2020a) on Colored-MNIST, WaterBirds, CelebA, MultiNLI, datasets, we pick the best learning rate parameter from $\{1e\text{-}3, 1e\text{-}5\}$, weight decay from $\{1e\text{-}4, .1, 1\}$, use SGD optimizer, and set the batch size to 1024, 128, 64, 32

---

[1]https://github.com/p-lambda/wilds

respectively. On WILDS datasets, we adopt the default optimization parameters[2], and only tune the step-size parameter.

We report the test set's performance corresponding to the best hyperparameter(s) and epoch on the validation split. We use standard train-validation-test splits for all the datasets when available.

**Backbone Model**

We followed closely the training setup of Sagawa et al. (2020a); Koh et al. (2020). On WaterBirds, CelebA dataset, we used ResNet50; on Colored-MNIST, PovertyMap, we used ResNet18; on Cemlyon17, FMoW, we used DenseNet121. All the image models except Colored-MNIST (which trains on $28 \times 28$ images), PovertyMap (which deals with multi-spectral version of ResNet18), are pretrained on ImageNet. MultiNLI, CivilComments, use a pretrained uncased DistilBERT-base model.

## 5.3 RESULTS

Table 4 shows the worst group test split accuracy for the four standard (non-WILDS) sub-population shift datasets. For all the tasks shown, ERM performs worse than a random baseline on the worst group, although the average accuracy is high. ERM-UW is a strong baseline, and improves the worst group accuracy of ERM on three of the four tasks, without hurting much the average accuracy. PGI is no better than ERM or ERM-UW on all the tasks except on Colored-MNIST. Group-DRO improves worst accuracy on most datasets, however, CGD fares even better. CGD improves worst-group accuracy on all the datasets over ERM. Except on MultiNLI text task, the gains of CGD are significant over other methods, and on MultiNLI task the worst group accuracy of CGD is at least as good as Group-DRO. On Colored-MNIST, which has the highest ratio of majority to minority group size, the gains of CGD are particularly large.

We report comparisons using the four WILDS datasets in Table 5. We show both in-domain (ID) and out-domain (OOD) generalization performance when appropriate, they are marked in the third row. All the results are averaged over multiple runs; FMoW numbers are averaged over three seeds, CivilComments over five seeds, Camelyon17 over ten seeds, and PovertyMap over the five data folds. Confirming with the WILDS standard, we report worst group accuracy for FMoW, Civil-Comments, worst region Pearson correlation for PovertyMap, average out-of-domain accuracy for Camelyon17. We make the following observations from the results. ERM is surprisingly strong on all the tasks except CivilComments. Strikingly, Group-DRO is worse than ERM on four of the five tasks shown in the table, including the in-domain (sub-population shift) evaluation on PovertyMap task. CGD is the only algorithm that performs consistently well across all the tasks. The results suggest CGD is significantly robust to sub-population shift, and performs no worse than ERM on domain shifts. Further, we study Colored-MNIST dataset under varying ratio of majority to minority group sizes in Appendix G and demonstrate that CGD is robust to sub-population shifts even under extreme training population disparity.

| Alg. ↓ | CMNIST | WaterBirds | CelebA | MultiNLI |
|---|---|---|---|---|
| ERM | 50 (0.0), 0 (0.0) | 85.2 (0.8), 61.2 (1.4) | **95.2 (0.2)**, 45.2 (0.9) | **81.8 (0.4)**, 69.0 (1.2) |
| ERM-UW | 53.6 (0.1), 7.2 (2.1) | **91.8 (0.2)**, 86.9 (0.5) | 92.8 (0.1), 84.3 (0.7) | 81.2 (0.1), 64.8 (1.6) |
| PGI | 52.5 (1.7), 44.4 (1.5) | 89.0 (1.8), 85.8 (1.8) | 92.9 (0.2), 83.0 (1.3) | 80.8 (0.8), 69.0 (3.3) |
| G-DRO | 73.5 (5.3), 48.5 (11.5) | 90.1 (0.5), 85.0 (1.5) | 92.9 (0.3), 86.1 (0.9) | 81.5 (0.1), **76.6 (0.5)** |
| CGD | 78.6 (0.5), **65.6 (5.9)** | 91.3 (0.6), **88.9 (0.8)** | 92.5 (0.2), **90.0 (0.8)** | 81.3 (0.2), **76.1 (1.5)** |

Table 4: Average and worst-group accuracy in that order. All numbers are averaged over 3 seeds and standard deviation is shown in parenthesis. G-DRO is abbreviation for Group-DRO.

## 6 RELATED WORK

Distributionally robust optimization (DRO) methods Duchi & Namkoong (2018) seek to provide uniform performance across all examples, through focus on high loss groups. As a result, they are not robust to the presence of outliers (Hashimoto et al., 2018; Hu et al., 2018; Zhai et al., 2021). Zhai et al. (2021) handles the outlier problem of DRO by isolating examples with high train loss.

---

[2]WILDS dataset parameter configuration Github URL.

| Algorithm↓ | Camelyon17 | PovertyMap | | FMoW | CivilComments |
|---|---|---|---|---|---|
| Metric → | Avg. Acc. | Worst Pearson r | | Worst-region Acc | Worst-Group Acc. |
| Eval. type → | OOD | ID | OOD | OOD | ID |
| CORAL | 59.5 (7.7) | **0.59 (0.03)** | **0.44 (0.06)** | 31.7 (1.2) | 65.6 (1.3) |
| IRM | 64.2 (8.1) | 0.57 (0.08) | 0.43 (0.07) | 30.0 (1.4) | 66.3 (2.1) |
| ERM | **70.3 (6.4)** | 0.57 (0.07) | **0.45 (0.06)** | **32.3 (1.2)** | 56.0 (3.6) |
| Group-DRO | 68.4 (7.3) | 0.54 (0.11) | 0.39 (0.06) | 30.8 (0.8) | **70.0 (2.0)** |
| CGD | **69.4 (7.8)** | **0.58 (0.05)** | 0.43 (0.03) | **32.0 (2.2)** | **69.1 (1.9)** |

Table 5: Evaluation on WILDS datasets: All numbers averaged over multiple runs, standard deviation is shown in parenthesis. Second row shows the evaluation metric, and the third shows the evaluation type: in-domain (ID) or out-of-domain (OOD). Two highest absolute performance numbers are marked in bold in each column.

Sagawa et al. (2020a) extend DRO to the case where training data comprises of a set of groups like in our setting. Dagaev et al. (2021); Liu et al. (2021); Creager et al. (2021); Ahmed et al. (2021) extend the sub-population shift problem to the case when the group annotations are unknown. They proceed by first inferring the latent groups of examples that negatively interfere in learning followed by robust optimization with the identified groups. In the same vein, Zhou et al. (2021); Bao et al. (2021) build on Group-DRO for the case when the supervised group annotations cannot recover the ideal distribution with no spurious features in the family of training distributions they represent. Zhou et al. (2021) maintains per-group and per-example learning weights, in order to handle noisy group annotations. Bao et al. (2021) uses environment specific classifiers to identify example groupings with varying spurious feature correlation, followed by Group-DRO's worst-case risk minimization.

Goel et al. (2020) augment the minority group with generated examples. However, generating representative examples may not be easy for all tasks. In contrast, Sagawa et al. (2020b) propose to sub-sample the majority group to match the minority group. In the general case, with more than two groups and when group skew is large, such sub-sampling could lead to severe data loss and poor performance for the majority group. Menon et al. (2021) partially get around this limitation by pre-training on the majority group first. Ahmed et al. (2021) considered as a baseline a modified version of Group-DRO such that all the label classes have equal training representation. However, these methods have only been applied on a restricted set of image datasets with two groups, and do not consistently outperform Group-DRO. In contrast, we show consistent gains over DRO and ERM up weighing, and our experiments are over eight datasets spanning image and text modalities.

**Domain Generalization** (DG) algorithms (Sun & Saenko, 2016; Shankar et al., 2018; Arjovsky et al., 2019; Ahmed et al., 2021) train such that they generalize to all domains including the seen domains that is of interest in the sub-population shift problem. However, due to the nature of DG benchmarks, the DG algorithms are not evaluated on cases when the domain sizes are heavily disproportionate such as in the case of sub-population shift benchmarks (Table 3). We compare with two popular DG methods: CORAL Sun & Saenko (2016) and IRM Arjovsky et al. (2019) on the WILDS benchmark, and obtain significantly better results than both.

## 7 CONCLUSION

Motivated by the need for modeling inter-group interactions for minority group generalization, we present a simple, new algorithm CGD for training with group annotations. We demonstrated the qualitative as well as empirical effectiveness of CGD over existing and relevant baselines through extensive evaluation using simple and real-world datasets. We also prove that CGD converges to FOSP of ERM objective even though it is not performing gradient descent on ERM.
**Limitations and Future work:** CGD critically depends on group annotated training data to remove spurious correlations or reduce worst case error on a sub-population. This limits the broad applicability of the algorithm. As part of future work, we plan to extend CGD to work in settings with noisy or unavailable group annotations following recent work that extends Group-DRO to such settings. The merits of CGD, rooted in modeling inter-group interactions, could be more generally applied to example weighting through modeling of inter-example interactions, which could be of potential leverage in training robust models without requiring group annotations.

## 8 REPRODUCIBILITY STATEMENT

We release the anonymized implementation of our algorithms publicly at this link, with instructions on running the code and pointers to datasets. We describe all our datasets in Section 5 and Appendix E. In Section 5, we also detail the hyperameter ranges, strategy for picking the best model. The detailed proof of our convergence theorem can be found in Appendix B.

## 9 ACKNOWLEDGEMENTS

The first author is supported by Google PhD Fellowship. We acknowledge engaging and insightful discussions with Divyat Mahajan and Amit Sharma during the initial phases of the project. We gratefully acknowledge the Google's TPU research grant that accelerated our experiments.

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

# Appendix

## A  SCALING RULE FOR GRADIENTS USING LOCAL QUADRATIC APPROXIMATION

The gradient norms can vary widely even for a given task depending on where we are in the loss landscape. We will now illustrate this issue. Consider any group $i$. Since the training loss $\ell_i(\theta) \geq 0$ for all $\theta$, any $\hat{\theta}$ satisfying $\ell_i(\hat{\theta}) = 0$ is an approximate global minimizer. For any such global minimizer with $\ell_i(\hat{\theta}) = 0$ and $\|\nabla \ell_i(\hat{\theta})\| = 0$, the local approximation of $\ell_i$ given by the second order Taylor expansion at $\hat{\theta}$ is:

$$\ell_i(\theta) \approx \frac{1}{2} \langle \theta - \hat{\theta}, \nabla^2 \ell_i(\hat{\theta}) \left( \theta - \hat{\theta} \right) \rangle \text{ and } \nabla \ell_i(\theta) \approx \nabla^2 \ell_i(\hat{\theta}) \left( \theta - \hat{\theta} \right).$$

Denoting $\underline{\sigma} = \sigma_{\min}(\nabla^2 \ell_i(\hat{\theta})) \geq 0$ and $\overline{\sigma} = \sigma_{\max}(\nabla^2 \ell_i(\hat{\theta})) \geq 0$ to be the smallest and the largest eigenvalue of $\nabla^2 \ell_i(\hat{\theta})$, we have that $\|\nabla \ell_i(\theta)\|$ bounded between $\sqrt{\underline{\sigma}} \cdot \sqrt{\ell_i(\theta)}$ and $\sqrt{\overline{\sigma}} \cdot \sqrt{\ell_i(\theta)}$. Consequently, the gradient norm can vary widely between smallest and largest eigenvalues. Besides, the gradient scale also depends on the number of parameters, task, dataset and architecture.

To summarize, we show that gradients are of the order of the loss, but can take a large spectrum of values, which makes the $\alpha$ updates unstable and $\eta_\alpha$ tuning difficult if used as is in equation 3. Since we are only interested in capturing if a group transfers positively or negatively, we retain the cosine-similarity of the gradient dot-product but control their scale through $\ell(\theta)^p$, for some $p > 0$. That is, we set the gradient $\nabla \ell_i(\theta)$ to $\frac{\nabla \ell_i(\theta)}{\|\nabla \ell_i(\theta)\|} \ell_i(\theta)^p$ leading to the final $\alpha$ update shown in equation 4.

When we set $p$ to a very large value, the gradient inner products $\ell_i{}^p \ell_j{}^p \cos(\nabla \ell_i(\theta), \nabla \ell_j(\theta))$, and hence the $\alpha$ value are largely influenced by the loss values. On the other hand, when we set $p$ to 0, we could get stuck in picking low loss train groups repeatedly without significant parameter update and hence not converge. In practice, neither of the extremes are ideal. We tried $p = \{0.25, 0.5, 1, 2\}$ on our synthetic setup, WaterBirds, CelebA, and found $p = 0.5$ to perform well empirically.

## B  PROOF OF THEOREM 1

*Proof of Theorem 1.* Let us consider the $t^{\text{th}}$ iteration where the iterate is $\theta^t$ and mixing weights are $\alpha^t$. Using the notation in Section 2, we denote $\mathcal{R}(\theta) = \frac{1}{k} \sum_i \ell_i(\theta)$, $g_i = \nabla \ell_i(\theta^t)$ and $g = \frac{1}{k} \sum_i g_i$. The update of our algorithm is given by:

$$\alpha_i^{t+1} = \frac{\alpha_i^t \cdot \exp\left(\eta_\alpha \langle g_i, g \rangle\right)}{Z} \tag{7}$$

$$\theta^{t+1} = \theta^t - \eta \sum_i \alpha_i^{t+1} g_i, \tag{8}$$

where $Z = \sum_j \alpha_j^t \cdot \exp\left(\eta_\alpha \langle g_j, g \rangle\right)$. Let us fix $\alpha^* = (1/k, \cdots, 1/k) \in \mathbb{R}^k$ and use $KL(p, q) = \sum_i p_i \log \frac{p_i}{q_i}$ to denote the KL-divergence between $p$ and $q$. We first note that the update equation 7 on $\alpha$ corresponds to mirror descent steps on the function $\langle g, \sum_i \alpha_i g_i \rangle$ and obtain:

$$
\begin{aligned}
KL(\alpha^*, \alpha^{t+1}) &= \sum_i \alpha_i^* \log \frac{\alpha_i^*}{\alpha_i^{t+1}} = \sum_i \alpha_i^* \log \frac{\alpha_i^* Z}{\alpha_i^t \cdot \exp\left(\eta_\alpha \langle g_i, g \rangle\right)} \\
&= \sum_i \alpha_i^* \log \frac{\alpha_i^*}{\alpha_i^t} - \eta_\alpha \alpha_i^* \langle g_i, g \rangle + \alpha_i^* \log Z \\
&= KL(\alpha^*, \alpha^t) - \eta_\alpha \|g\|^2 + \log \left( \sum_i \alpha_i^t \exp\left(\eta_\alpha \langle g_i, g \rangle\right) \right).
\end{aligned}
\tag{9}
$$

Since each $\ell_i(\cdot)$ is $G$-Lipschitz, we note that $\|g_i\| \leq G$ for every $i$. Consequently, $\|g\| \leq G$ and $|\langle g_i, g \rangle| \leq G^2$. Using the fact that $\exp(x) \leq 1 + x + x^2$ for $|x| < 1$, and since $\eta_\alpha \leq \frac{1}{G^2}$, we see that:

$$\log\left(\sum_i \alpha_i^t \exp\left(\eta_\alpha \langle g_i, g \rangle\right)\right) \leq \log\left(\sum_i \alpha_i^t \left(1 + \eta_\alpha \langle g_i, g \rangle + \left(\eta_\alpha G^2\right)^2\right)\right)$$

$$= \log\left(1 + \left(\sum_i \alpha_i^t \eta_\alpha \langle g_i, g \rangle\right) + \left(\eta_\alpha G^2\right)^2\right)$$

$$\leq \left(\sum_i \alpha_i^t \eta_\alpha \langle g_i, g \rangle\right) + \left(\eta_\alpha G^2\right)^2,$$

where we used $\log(1 + x) \leq x$ in the last step. Plugging this back into equation 9, we obtain:

$$KL(\alpha^*, \alpha^{t+1}) \leq KL(\alpha^*, \alpha^t) + \left(\sum_i \alpha_i^t \eta_\alpha \langle g_i, g \rangle\right) + \left(\eta_\alpha G^2\right)^2 - \eta_\alpha \|g\|^2$$

$$\Rightarrow -\sum_i \alpha_i^t \langle g_i, g \rangle \leq -\|g\|^2 + \frac{KL(\alpha^*, \alpha^t) - KL(\alpha^*, \alpha^{t+1})}{\eta_\alpha} + \eta_\alpha G^4. \tag{10}$$

We will now argue that $\sum_i \alpha_i^t \langle g_i, g \rangle \leq \sum_i \alpha_i^{t+1} \langle g_i, g \rangle$. To show this, it suffices to show that:

$$\sum_i \alpha_i^t \langle g_i, g \rangle \leq \sum_i \alpha_i^{t+1} \langle g_i, g \rangle$$

$$\Leftrightarrow \left(\sum_i \alpha_i^t \langle g_i, g \rangle\right)\left(\sum_i \alpha_i^t \exp\left(\eta_\alpha \langle g_i, g \rangle\right)\right) \leq \sum_i \alpha_i^t \langle g_i, g \rangle \exp\left(\eta_\alpha \langle g_i, g \rangle\right)$$

$$\Leftrightarrow \sum_i \alpha_i^t \left(\langle g_i, g \rangle - \overline{ip}\right)\left(\exp\left(\eta_\alpha \langle g_i, g \rangle\right) - \overline{exp}\right) \geq 0,$$

where $\overline{ip} = \sum_i \alpha_i^t \langle g_i, g \rangle$ and $\overline{exp} = \sum_i \alpha_i^t \langle g_i, g \rangle \exp\left(\eta_\alpha \langle g_i, g \rangle\right)$. The last inequality follows from the fact that $\exp(\cdot)$ is an increasing function and hence for any random variable $X$, covariance of $X$ and $\exp(X)$ is greater than or equal to zero.

Now coming back to the update equation 8, we can argue its descent property as follows:

$$\mathcal{R}(\theta^{t+1}) \leq \mathcal{R}(\theta^t) - \eta \langle g, \sum_i \alpha_i^{t+1} g_i \rangle + \frac{\eta^2 L}{2} \|\sum_i \alpha_i^{t+1} g_i\|^2$$

$$\leq \mathcal{R}(\theta^t) + \eta\left(-\|g\|^2 + \frac{KL(\alpha^*, \alpha^t) - KL(\alpha^*, \alpha^{t+1})}{\eta_\alpha} + \eta_\alpha G^4\right) + \frac{\eta^2 L}{2} \|\sum_i \alpha_i^{t+1} g_i\|^2$$

$$\leq \mathcal{R}(\theta^t) + \eta\left(-\|g\|^2 + \frac{KL(\alpha^*, \alpha^t) - KL(\alpha^*, \alpha^{t+1})}{\eta_\alpha} + \eta_\alpha G^4\right) + \frac{\eta^2 L G^2}{2}$$

$$\Rightarrow \|g\|^2 \leq \frac{\mathcal{R}(\theta^t) - \mathcal{R}(\theta^{t+1})}{\eta} + \frac{KL(\alpha^*, \alpha^t) - KL(\alpha^*, \alpha^{t+1})}{\eta_\alpha} + \eta_\alpha G^4 + \frac{\eta L G^2}{2}.$$

Summing the above inequality over timesteps $t = 1, \cdots, T$, we obtain:

$$\frac{1}{T}\sum_{t=0}^{T-1} \|\nabla \mathcal{R}(\theta^t)\|^2 \leq \frac{\mathcal{R}(\theta^0) - \mathcal{R}(\theta^T)}{\eta T} + \frac{KL(\alpha^*, \alpha^0) - KL(\alpha^*, \alpha^T)}{\eta_\alpha T} + \eta_\alpha G^4 + \frac{\eta L G^2}{2}$$

$$\leq \frac{2B}{\eta T} + \eta_\alpha G^4 + \frac{\eta L G^2}{2},$$

where we used $KL(\alpha^*, \alpha^{t+1}) \geq 0$, $KL(\alpha^*, \alpha^0) = 0$ and $\mathcal{R}(\theta^0) - \mathcal{R}(\theta^T) \leq 2B$ in the last step. Using the parameter choices $\eta = 2\sqrt{\frac{B}{LG^2 T}}$ and $\eta_\alpha = \sqrt{\frac{BL}{G^6 T}}$, we obtain that:

$$\frac{1}{T}\sum_{t=0}^{T-1} \|\nabla \mathcal{R}(\theta^t)\|^2 \leq 3\sqrt{\frac{BLG^2}{T}}.$$

This proves the theorem. $\qquad \square$

## C    More details of the synthetic settings

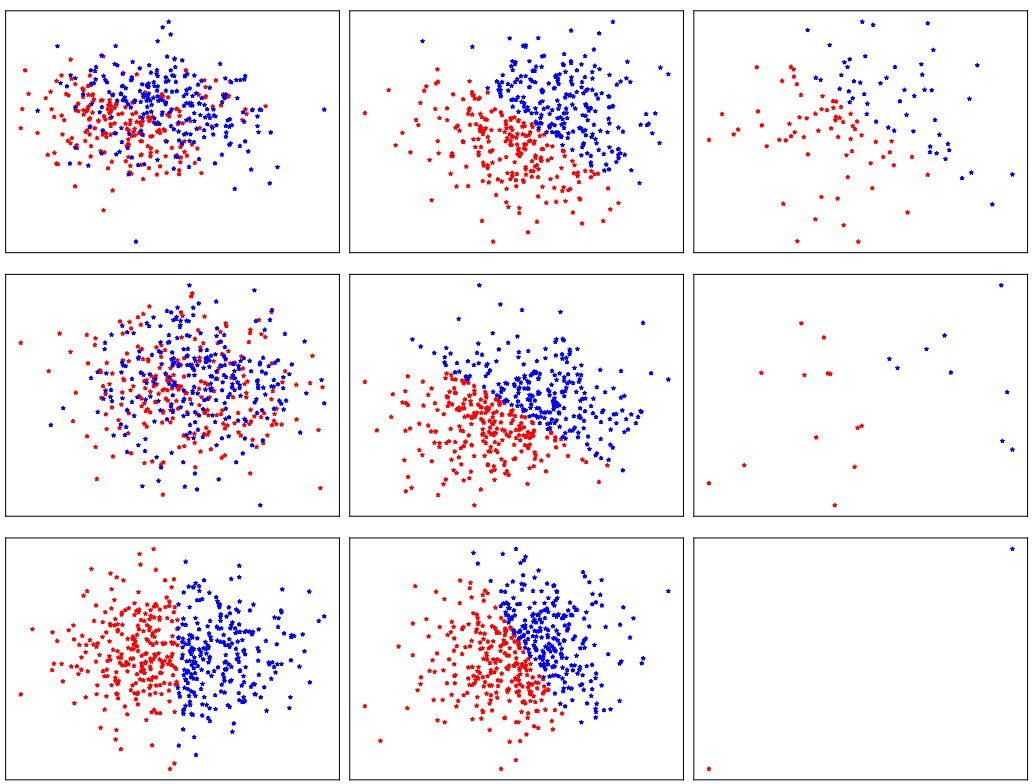

Table 6: Plot of data from first two features for the three synthetic settings. The three columns show the distribution of examples from the three groups in the order. The rows from top to bottom show the plots for Noisy (Section 4.1), Spurious (Section 4.3) and Rotation (Section 4.2) simple settings. The minority group for Rotation setting (right bottom) has only two examples. We omitted from the plot the third feature of the spurious setting (second row).

In Table 6, we show the data for each group and each simple setting we considered in Section 4. In the noise setting of the first row, the noise is only introduced in one of the groups. For the spurious case of second row, the first two features perform poorly on the first group while the third spurious feature (not shown in the figure) performs poorly on both the second and the third group. Shown in the last row, the classifier is gradually rotated for the rotation case and the minority group is extremely sparse with only two examples.

## D    Qualitative experiments with deep models

In this section, we repeat the qualitative experiments of Section 4 with MNIST examples and deep models. For all the setting below, we train a ResNet-18 model with standard SGD, pick the best hyperparameter based on the validation set, and report the performance on the test set corresponding to the checkpoint performing best on the validation set. We binarized the label space by grouping the first five digits in to one class and the next five in to another. In all cases, we create three groups of examples by randomly partitioning the training set into 4,900, 4,900, and 200 examples. We illustrate various aspects of the differences between Group-DRO and CGD by perturbing instances in a group in three different ways below:

### D.1    Label Noise setup

In this case, we randomly flipped all the labels in the first group, meaning the first group is highly noisy in the training data while the test/validation set is noise-free. We trained a ResNet-18 model

on all the three groups with a parameter weight decay of 1. Since CGD focuses on high transfer groups instead of high loss groups, it is more robust to the presence of a noisy training group when compared with Group-DRO. As a result, a model trained with CGD has better worst group accuracy when compared with Group-DRO (Noise-MNIST column of Table 2).

### D.2 UNEVEN INTER-GROUP SIMILARITY

In this case, we rotate digits in the second and third groups by 30, 60 degrees. The difference in degree of rotation quantifies the transferability between the groups. The second group transfers better to all the groups compared to the first and third. CGD, by design, focuses training on the high-transfer second group more than first and performs better on all the rotations when compared with Group-DRO (Rotation-MNIST column of Table 2).

### D.3 SPURIOUS CORRELATIONS SETUP

Here, on three group setup, we use a color feature to spuriously correlate with label in the first and last group, but not in the second group as follows: Digits in the first group are colored red for label 0 and blue for label 1, colored blue or red randomly in the second group, and colored blue for label 0 and red for label 1 in the third group. CGD performs better than Group-DRO by focusing mostly on the clean and high transfer second group. As a result, generalizes better than Group-DRO (Spurious-MNIST column of Table 2)

## E  DATASETS

**Colored-MNIST:** Using MNIST (LeCun & Cortes, 2010) digits, we create a dataset where the foreground color is spuriously correlated with the label. We split 50,000 examples into majority, minority with 1000:1 group size ratio, and binarize the label space to predict digits 0-4, 5-9 as the two classes. Test data has equal proportion of each group. In the majority group, the foreground is red for examples of label 0 and blue for label 1; the minority group examples were colored in reverse. Color based prediction of the label cannot generalize, however, color is spuriously and strongly correlated with the label in the training data. Many previous work (Arjovsky et al., 2019; Ahmed et al., 2021) adopted Colored MNIST for probing generalization.

**WaterBirds** task is to predict if an image contains water-bird or a land-bird and contains two groups: majority, minority in ratio 62:1 in training data. The minority group has reverse background-foreground coupling compared to the majority. Test/validation data has equal proportion of each group.

**CelebA** (Liu et al., 2018) task is to classify a portrait image of a celebrity as blonde/non-blonde, examples are grouped based on the gender of the portrait's subject. The training data has only 1,387 male blonde examples compared to 200K total examples. Test/validation data follows same group-class distribution as train.

**MultiNLI** (Williams et al., 2017) task is to classify a pair of sentences as one of *entailment, neutral, contradiction*. Gururangan et al. (2018) identified that negation words in the second sentence are spuriously correlated with the contradiction labels. Accordingly, examples are grouped based on if the second sentence contains any negation word. Test/validation data follows same group-class distribution as train.

**CivilComments-WILDS** (Borkan et al., 2019) task is to classify comments as toxic/non-toxic. Examples come with eight demographic annotations: *white, black, gay, muslim*, based on if the comment mentioned terms that are related to the demographic. Label distribution varies across demographics. In the bechmark train examples are grouped on *black* demographic, and testing is on the worst group accuracy among all 16 combinations of the binary class label and eight demographics.

**PovertyMap-WILDS** (Yeh et al., 2020) task is to classify satellite images in to a real valued wealth index of the region. The rural and urban sub-population from different countries make the different train groups. Bechmark evaluates on worst-region Pearson correlation between predicted and true wealth index in two settings: An in-domain setting that measures sub-population shift to seen regions, and an out-domain setting that evaluates generalization to new countries.

**FMoW-WILDS** (Christie et al., 2018) task is to classify RGB satellite images to one of 62 land use categories. Land usage differs across countries and evolves over years. Training examples are stratified in to eleven groups based on the year of satellite image acquisition. We report out-domain evaluation using test data from regions of seen countries but from later years, and measures worst accuracy among the five geographical regions: Africa, Americas, Oceania, Asia, and Europe. The in-domain evaluation with worst accuracy on the eleven years is unavailable for other algorithms, so we stick to only out-domain evaluation.

**Camelyon17-WILDS** (Bandi et al., 2018) is a binary classification task of predicting from a microscope image of a tissue if it contains a tumour. The training data contains scans from three hospitals, and the test data contains scans from multiple unseen hospitals. The evaluation metric is average accuracy on the test set hospitals.

## F  MORE TECHNICAL DETAILS

**Predictive Group Invariance** (Ahmed et al., 2021) proposed an algorithm that was shown to generalize to various kinds of shifts including in-distribution shifts. The algorithm penalizes discrepancy in predictive probability distribution across groups with the same label as shown below.

$$\mathcal{R}(\theta) = \sum_i \frac{n_i}{\sum_i n_i} \ell_i + \lambda \sum_c \mathbb{E}_{Q^c}[p_\theta(y \mid x)] \log \frac{\mathbb{E}_{Q^c}[p_\theta(y \mid x)]}{\mathbb{E}_{P^c}[p_\theta(y \mid x)]}$$

Where 'c' is the class label and the rest of terms in the equation follow the notation of this paper. For a given class label 'c', the distributions: $Q^c, P^c$, represent the examples of the same label but belong to two different inferred groups. They used an existing work (Creager et al., 2021) to stratify the examples in to groups. Here, we used the available group annotations for creating the example splits: $Q^c, P^c$, for a fair comparison. We picked the value of $\lambda$ from {1e-3, 1e-2, 1e-1, 1}.

The original paper noted that the second distribution of the KL divergence is an "easy" group such that its average predictive distribution is mostly correct. In our case, since all the groups contain bias, they are all "easy", and hence the direction of divergence is unclear. Since ERM tends to learn the biases aligned with the majority group, we set the $P^c$ distribution from the majority and $Q^c$ from the minority.

**CGD**: We model inter-group transfer characteristics using inner products of first order gradients (equation 4). Since per-group gradient computation for all the parameters is computationally expensive, we use only a subset of parameters. We use only the last three layers for ResNet-50, Densenet-101 and DistilBERT, and all the parameters for gradient computation for any other network.

## G  PERFORMANCE AT VARYING LEVELS OF HETEROGENEITY

We study performance on the simple synthetic setting of Colored-MNIST under varying levels of heterogeneity, i.e. ratio of majority to the minority group. As discussed in the datasets section, ColoredMNIST dataset has two groups: a majority group where the label and the digit's color match and the minority group where they do not match. The examples are grouped in to majority and minority in the ratio of r to 1. As the value of r increases, the net (spurious) correlation of the color with the label increases, furthering the gap between the best and worst groups. The train split has 50,000 examples and validation, test split have 10,000 examples each. The validation, test split contain all the groups in equal proportions. We follow the same optimization procedure as de-

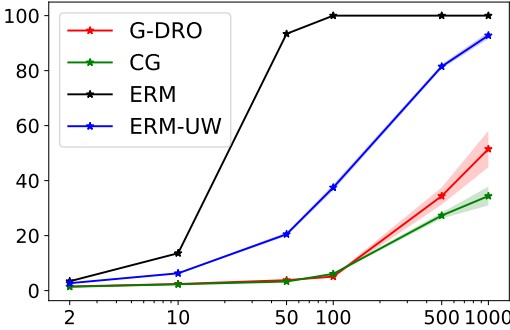

Figure 6: Worst group error rate with increasing ratio of majority to minority. Shaded region shows one standard error. All estimates are aggregated over three runs.

scribed for the ColoredMNIST as described in the main content, with the exception that we set the weight decay to 0.

We trained and evaluated different algorithms for the ratio (r) going from 2 to 1000. Figure 6 shows the worst-group error rate with the value of r for various algorithms. CGD performed well across all the values of r, and for the extreme value of r=1000: CGD's worst error rate of 34.4% is far better than the 51.5% of Group-DRO.

## H  HOW DOES SYNTHETIC EXPERIMENTS RELATE TO REAL-WORLD DATASETS?

In this section we discuss the similarities between our synthetic settings of Section 4 and the real-world datasets 5.1.

The text benchmarks (MultiNLI, CivilComments-WILDS) and CelebA resemble our toy setup of Sec. 4.3. In MultiNLI, the examples with negation words have spurious correlations that negatively transfer between the contradiction and entailment labels. The examples from the group with no negation words do not contain such spurious features. Similarly, in CivilComments-WILDS the examples from the black group contain spurious correlation (they contain tokens that identify the demographic, which can be easily exploited to classify examples from black group as mostly toxic), while such spurious features are absent in the non-black group. In the CelebA dataset too, male non-blonde (majority) negatively transfers to the male blonde (minority) since the classifier may learn to interpret short hair (male) to be non-blonde. On the other hand, the female blond/non-blond groups do not contain any known spurious correlation. In all the cases, CGD could avoid learning spurious features by focussing training on groups with no or relatively low spurious correlation similar to what was demonstrated in Sec 4.3, thereby learning a more robust solution with dampened strength of spurious features.

FMoW-WILDS, PovertyMap-WILDS are similar to our label noise simple setup of Sec. 4.1. FMoW-WILDS task is to classify a satellite image into one of 62 land-use categories. The dataset is annotated with human curators labeling if a marked region in an image contains, say a "police station" (Yeh et al., 2020). Depending on the demographic spread of the human curators, the label correctness is expected to vary from one region to another. Also, some land-use categories are far easier to classify than others (for eg. "police station" vs "helipad"). Similarly, PovertyMap-WILDS task is to map a satellite image to its poverty index. Poverty index per region (urban/rural settlement) ground-truth was acquired through secondary sources such as asset index of the region from the national demographic surveys (Christie et al., 2018). The asset to wealth index per region was found to vary per country and hence the quality of the label. The non-uniform label noise of the two datasets is similar to our setup in Sec. 4.1. CGD focuses only on the difficult groups that transfer well to the rest, unlike Group-DRO that only pursue the maximum loss groups.

