# OpenReview forum: "Focus on the Common Good: Group Distributional Robustness Follows"
_ICLR.cc/2022/Conference — ICLR 2022 Poster_

### Official Review · Reviewer_UQv8 · 2021-11-01

**Correctness:** 3
**Technical Novelty And Significance:** 4
**Empirical Novelty And Significance:** 4
**Recommendation:** 8
**Confidence:** 4

**Main Review:**

The proposed method is easily understood and implemented. It also has good performance empirically. The writing is clear and the intuition behind the method is clearly conveyed in the simulation study. The intuition of inter-group interactions seems reasonable to me.

Weakness:
For the detailed algorithm, There is somewhere that is unclear to me.
1. Close to convergence, the statement that the gradient $\nabla ||\ell_i( \theta_t)||$ can be much smaller than the loss value $\ell_i(\theta_t)$ is unclear to me. In fact, consider $\ell(\theta) = ||\theta ||^2_2$. It should be $\ell(\theta_t) << ||\nabla \ell(\theta_t)||$ when close to the convergence. In Appendix A, why ||nabla \ell(\theta_t)|| could be as small as $\sqrt{\underline{\sigma} \ell_i(\theta)}$. And why is $\underline{\sigma}$ close to zero?

2. When $<g_i,g_j> = 0$, it does not exactly match the group-DRO update steps mentioned at the bottom of Page 2, since there is regularization between consecutive time steps.
3. Please formally define $cos(g_i,g_j)$.
3. At the end of paper 3, the author mentioned: " The corrected loss values simply replace the $\ell_i$ of Line 5 in Algorithm 1". However, the correction $\ell_i = \ell_i + C/\sqrt{n}$ does affect $\nabla \ell_i(\theta)$. I think the author should refer to updating step (4).

For the simulations and experiments:
1. At Page 6, "y, 1-y on the third group, and set to y or 1-y w.p. 0.6" should be set to y w.p 0.6
2. For the datasets, what are population types "Label * Group" or Group? Why does the worst ratio refect loosely the strength of the spurious correlation?

**Summary Of The Paper:**

The paper provides an efficient method to generalize to all groups in the presence of sub-population shifts and domain adaptation. The paper conducts extensive simulations to derive insights and also numerical experiments on the benchmark dataset to demonstrate the performance. The proposed method is intuitive, easily implemented, and has good performance.

**Summary Of The Review:**

This paper provides a practical algorithm to the broad field of distributional shifts, domain adaptation, and fairness.

---

> ### Author Response · Authors · 2021-11-12
> **Response to questions**
>
> Thanks for the feedback and suggestions, we answer your questions below.
>
> > Close to convergence, the statement that the gradient ...
>
> The claim that $||\nabla \ell_i(\theta_t)||$ can be much smaller than the loss value $ \ell_i(\theta_t)$ is indeed a typo --  indeed $||\nabla \ell_i(\theta_t)||$ can be of order $\sqrt{\ell_i(\theta_t)}$ and consequently $||\nabla \ell_i(\theta_t)|| >> \ell_i(\theta_t)$. What we actually meant to say was that when $\ell_i(\theta_t) \rightarrow 0$, we also have $||\nabla \ell_i(\theta_t)|| \rightarrow 0$.
> $||\nabla \ell_i(\theta_t)|| \gtrsim \sqrt{\underline{\sigma}\ell_i(\theta_t)}$ can be seen as follows:
>
> $||\nabla \ell(\theta)||^2 \approx (\theta - \hat{\theta})^\top \left(\nabla^2 \ell(\hat{\theta}) \right)^2 (\theta - \hat{\theta}) = \left(\left(\nabla^2 \ell(\hat{\theta}) \right)^{1/2}(\theta - \hat{\theta})\right)^\top \left(\nabla^2 \ell(\hat{\theta}) \right) \left(\left(\nabla^2 \ell(\hat{\theta}) \right)^{1/2}(\theta - \hat{\theta})\right)$
>
> $\geq \underline{\sigma} ||\left(\nabla^2 \ell(\hat{\theta}) \right)^{1/2}(\theta - \hat{\theta})||^2 = \underline{\sigma} (\theta - \hat{\theta})^\top \nabla^2 \ell(\hat{\theta}) (\theta - \hat{\theta}) \approx \underline{\sigma} \cdot \ell(\theta)$
>
> Since the number of parameters in neural networks is often larger than the number of training examples, there are often zero or close to zero eigenvalues of the Hessian at a local minimum and has been observed empirically as well [1].
>
> [1] Empirical Analysis of the Hessian of Over-Parametrized Neural Networks by L. Sagun et al. 2017
>
> > For the datasets, what are population types "Label * Group" or Group? Why does the worst ratio reflect loosely the strength of the spurious correlation?
>
> Population type in the table shows what is used for creating the training sub-population. For eg., label x group population type means we create sub-population for every label, group pair.
>
> The average label correlation of the spurious feature is: majority size * $\rho$ + minority size * -$\rho$ for some correlation coefficient $\rho$ of the spurious feature with the label in the majority group. When majority size/minority size >> 1, then the average label correlation is far from being 0 and consequently much harder to avoid learning. This is why the worst ratio reflects the strength of the spurious correlation.

---

### Official Review · Reviewer_cTFe · 2021-11-02

**Correctness:** 3
**Technical Novelty And Significance:** 3
**Empirical Novelty And Significance:** 3
**Recommendation:** 6
**Confidence:** 3

**Main Review:**

PROS:

- Building an ERM-based method that performs well on the whole dataset without sacrificing the prediction accuracy on the group-level worse case really makes sense to me, especially when different groups have different amounts of label noise.
- The paper is well written, with a clear presentation style, and well-supported contributions, and easy to follow with good details on the experimental setup.
- The theoretical analyses and empirical understanding provide additional insight into this new algorithm.

Concerns:

- In some scenarios, the group annotations are available. Should we directly discard this information?
- What is the computational complexity of the algorithm?
- Is this method vulnerable to label noise?


*** Post Rebuttal: After reading the authors' response and the updated components of the manuscript, I thank the authors for addressing nearly all of my concerns. The inclusion of a clearer motivation, more discussion w.r.t. CGD and datasets, all enhance my understanding of the contributions of the paper beyond my original review. I decide to vote for acceptance. ***

**Summary Of The Paper:**

This paper proposes a novel ERM-based method for classification task with group annotated training data. The goal is to be group distributionally robust while enhancing the minority performance. The authors make an improvement to an existing method named Group-DRO by modifying the focus on the group with the highest regularized loss to focus on the group that leads to the largest decrease in average training loss. They analyze the convergence and present detailed comparisons with Group-DRO.

**Summary Of The Review:**

Overall, I think this is an interesting and solid paper.

---

> ### Author Response · Authors · 2021-11-12
> **Response questions and concerns**
>
> Thanks for the feedback, please find our responses below.
>
> > In some scenarios, the group annotations are available. Should we directly discard this information?
>
> The multi-group training problem assumes that we do not have access to group annotations during test-time (deployment), and hence cannot directly utilize them if available. However, algorithms that can exploit any available group annotations for even better classification would be interesting.
>
> > What is the computational complexity of the algorithm?
>
> Our method has the overhead of computing the gradient dot-products as shown in eq. 4, which is negligible compared to the cost of gradient computation. In this regard, our computational complexity is of the same order as Group-DRO or ERM.
>
> > Is this method vulnerable to label noise?
>
> CGD is much less vulnerable than GroupDRO to label noise as illustrated in Sec 4.1 toy setup.

---

### Official Review · Reviewer_4DaZ · 2021-11-02

**Correctness:** 3
**Technical Novelty And Significance:** 2
**Empirical Novelty And Significance:** 3
**Recommendation:** 3
**Confidence:** 4

**Main Review:**

Originality & Quality: While I like the simplicity of the proposed approach, there are a couple of glaring weaknesses in the paper:



1). The idea that we shouldn't minimize the worst-group error but rather minimize the error on the group that decreases all other groups' errors is intriguing! Definitely, the extant approach of minimizing the worst group error has generalization problems and hence doesn't perform best on all the benchmarks. However, the alternative proposed in the paper is not motivated, but rather comes across as an "ad-hoc" idea that "just seems to work". This is also corroborated by the fact that the proposed algorithm doesn't perform gradient descent on a particular loss function. That's the biggest weakness of this paper, especially in times when we want to infuse more *science* and *rigor* into deep learning. Sure, the paper tries to motivate the algorithm by saying that the "worst group" approach doesn't model inter-group similarity, but that intuition is nebulous at best.


2). The results on the synthetic datasets seem contrived, but that's OK as one can design synthetic dataset to make their point. However, the standard benchmarks shown in Table 3 are also altered. For example, the CivilComments dataset is shown as a 2 group when it is originally a 8-group task (the groups being the demographics of the users) as shown in the WILDS dataset paper. Similarly, the MultiNLI dataset contains 2-groups that seem contrived.


Clarity: The paper is well written and puts itself nicely in context of previous work. The overall presentation of the paper is good.

Significance: The paper addresses an important problem of robust ML and proposes a new method for improving the "group" robustness.


**Summary Of The Paper:**

The paper proposes a new method for robust ML under distribution shifts. Past work has looked at formulations that minimize the worst group error. This paper adds a new twist on it and instead argues for focusing on the group that leads to the greatest decrease in average training error for all the groups. This intuition is combined into an algorithm and the paper proves that though their proposed algorithm doesn't minimize a specific loss function, it still finds first-order-stationary points. The results are shown on several synthetic datasets as well as on the WILDS Robust ML benchmark that show the superior performance of the proposed algorithm over several baselines.



Main Contributions:

1). The paper proposes a new approach for robust ML under distribution shifts that performs gradient descent not on the group with worst error but on the group which decreases the average error of all other groups.

2). Results are shown on synthetic and real-world datasets which show the superior performance of the proposed method in achieving group robustness.


**Summary Of The Review:**

The paper provides an intriguing idea to improve group distributional robustness, but at its heart it is just an *ad-hoc algorithm* that *seems to work*. It is unclear what really is being optimized since there is no clear loss function that is minimized. The results also seem a little unconvincing since some of the real-world datasets are altered and not used in their original form, e.g., CivilComments, which raises questions regarding if the approach even works at all, irrespective of the issues regarding lack of theory behind the algorithm.

---

> ### Author Response · Authors · 2021-11-12
> **Response on our objective and real-world datasets**
>
> Thanks for the feedback, please find our response to your concerns below.
>
> > However, the alternative proposed in the paper is not motivated, but rather comes across as an "ad-hoc" idea that "just seems to work". This is also corroborated by the fact that the proposed algorithm doesn't perform gradient descent on a particular loss function.
>
> We apologize for our writing not making this point clear -- though our algorithm is not gradient descent (GD), it optimizes the macro-average loss function $\mathcal{R}(\theta)$. Theorem 1 shows that our algorithm monotonically decreases this macro average loss $\mathcal{R}(\theta)$ and eventually finds first order stationary points.
>
> From an optimization perspective, GD is only one of the many algorithms that can be used to optimize a loss function. Other examples are steepest descent, Newton’s method etc. While all of these algorithms are acceptable from an optimization viewpoint, they could differ significantly in terms of their generalization performance. The main benefit of our algorithm CGD over GD is that it finds a better generalizing solution.
>
> **Intuition behind why CGD finds a better generalizing solution than GD**: In the presence of spurious features in some groups, GD ends up relying on these spurious features and consequently does not generalize well on groups without these spurious features. By using gradients from those groups that yield benefits to all the groups, CGD promotes learning of features that are common across all groups, thereby obtaining better generalization on all the groups. This intuition has been successful in the related problem of domain generalization as well [1,2]. While there is enough empirical evidence to strongly support this intuition, rigorous theoretical evaluation of this intuition is still missing in literature and is an interesting open question. We will add this discussion to the paper.
>
> > However, the standard benchmarks shown in Table 3 are also altered. For example, the CivilComments dataset is shown as a 2 group when it is originally a 8-group task (the groups being the demographics of the users) as shown in the WILDS dataset paper. Similarly, the MultiNLI dataset contains 2-groups that seem contrived
>
> We did not alter the standard benchmarks.; Our experiment setup is exactly like GroupDRO evaluation on the CivilComments-WILDS dataset [3], GroupDRO only uses black/non-black (two groups) group supervision during training and report worst group accuracy among 16 sub-population (8 groups and 2 labels). Similarly, on MultiNLI, we exactly adopt the standard practice on the dataset of supervising with two group labels: whether a sentence contains any negation words or not [4]. We kept the data loaders, group supervision, optimization parameters, consistent with GroupDRO [3, 4] for a fair evaluation on all the datasets.
>
> References
> 1. Li, D., Yang, Y., Song, Y., and Hospedales, T. M., Deeper, broader and artier domain generalization. In IEEE International Conference on Computer Vision, ICCV, 2017
> 2. Piratla, V., Netrapalli, P. and Sarawagi, S., Efficient domain generalization via common-specific low-rank decomposition. In International Conference on Machine Learning, ICML 2020
> 3. Koh, P.W., Sagawa, S., Xie, S.M., Zhang, M., Balsubramani, A., Hu, W., Yasunaga, M., Phillips, R.L., Gao, I., Lee, T. and David, E., 2021, July. Wilds: A benchmark of in-the-wild distribution shifts. In International Conference on Machine Learning (pp. 5637-5664). PMLR
> 4. Sagawa, S., Koh, P.W., Hashimoto, T.B. and Liang, P., 2019. Distributionally robust neural networks for group shifts: On the importance of regularization for worst-case generalization. arXiv preprint arXiv:1911.08731.

---

> > ### Author Response · Authors · 2021-11-26
> > **Further clarification on CGD**
> >
> > Thanks for your response. Here is a simple Generative Model to demonstrate the rationale behind CGD.
> >
> > **Setting**:
> > For a group $i \in \{1,2,3\}$, and label y, let the data from group i be generated as $x = y(e_c + \beta_i e_s) + N(0, 1), \forall i \in [D]$ for some common and group-specific components $e_c, e_s$, and number of samples $n_i$ for group i satisfying $n_1 = n_2 >> n_3$. The values of $\beta_i$ are such that $\beta_1>\beta_2\approx 0>\beta_3$. Since a classifier that relies on $e_s$ helps majority groups at the expense of minority groups, we would like to learn a linear classifier that predominantly relies on $e_c$. The question then boils down to understanding which method can learn to ignore $e_s$ and rely mostly on $e_c$. This is similar to the data generation setup we considered in Section 4.3, with $e_c=[1, 1, 0]/\sqrt 2, e_s=[0, 0, 1]$, with $\beta=1, \approx 0, -1$ on the three groups.
> >
> > For each group i: the normalized negative gradient of a group 'i' has the form: $g_i = \lambda_i(e_c+\beta_i e_s+ \gamma_i n_i), n_i$ is a unit norm vector orthogonal to $e_c$ and $e_s$, representing the noise component due to finite samples and $\lambda_i = 1/\sqrt{1+ \beta_i^2 + \gamma_i^2}$ is the normalization factor. Given the number of samples, we note that $\gamma_1, \gamma_2 \approx 0$ and $\gamma_3 >> \gamma_1, \gamma_2$, and  $1>>\gamma_3^2$.
> >
> > **Gradient approximation at beginning of training**: We assume all the losses and gradient norms are equal at the beginning of training. We therefore analyze the CGD updates with the normalized gradients.
> >
> > **What ERM does**:
> > ERM, ERM-UW uses the gradient from all the groups, including the smallest group and consequently overfits on the smallest group.
> >
> > **What CGD does**:
> >  CGD obtains the gradient ($\sum_i \alpha^*_ig_i$) through $\alpha^* = \arg\max_\alpha <\sum_i \alpha_ig_i, \sum_i g_i>$, which with our form of gradients leads to:
> > With the shorthand notation: $a=[\lambda_i], b=[\lambda_i\beta_i], c=[\lambda_i^2 \gamma_i^2] $
> >
> > $\alpha^* = \arg\max_\alpha(\sum_i \lambda_i)(<\alpha, a>) +(\sum_i \lambda_i \beta_i)(<\alpha, b>) + <\alpha, c>$
> >
> > (using the fact that all components are unit norm and orthogonal)
> >
> > $= \arg\max_\alpha <\alpha, (\sum_i \lambda_i)a + (\sum_i \lambda_i \beta_i)b + c)>$
> >
> > The objective is maximized if $\alpha$ is in the direction of $(\sum_i \lambda_i)a + (\sum_i \lambda_i \beta_i)b + c$.
> >
> > Let us now plug in the values for our specific case of Section 4.3.
> > $\beta_1=1, \beta_2\approx 0, \beta_3=-1$.
> > $\lambda_i=\frac{1}{\sqrt {1+\beta_i^2+\gamma_i^2}}$, so $\lambda_i$ and $a_i$ are highest when $\beta_i$ and $\gamma_i$ are lowest (group 2).
> > $\sum_i\lambda_i\beta_i=1/\sqrt 2 + 0 - 1/\sqrt{2+\gamma_3^2}$ using $\gamma_1, \gamma_2\approx 0$. Also, $c=[0, 0, \gamma_3^2/(2+\gamma_3^2)]$.
> > In the expression for $\alpha^*$ above, both the second term (($\sum_i \lambda_i \beta_i)b$) and the third term (c) are $\approx$ 0 and can be ignored since $2/\gamma_3^2>>1$. In the left-over first term, the vector 'a' has larger support along the second group that has low $\beta, \gamma$, and hence, so does $\alpha^*$. This helps us understand why CGD picks the second group over the others. This is further corroborated by experiments in Figure 5.
> > CGD approximates the ERM-UW gradient without using the last group’s gradient and is therefore less prone to noise or overfitting.
> >
> > ---
> > More generally, although ERM-UW, CGD have the same objective function, their descent directions are different. While ERM-UW descends along the empirical average of group gradients, CGD descends along the common direction.

---

### Official Review · Reviewer_4c9A · 2021-11-03

**Correctness:** 4
**Technical Novelty And Significance:** 4
**Empirical Novelty And Significance:** 4
**Recommendation:** 6
**Confidence:** 4

**Main Review:**

Pros

- I feel the idea of training on the group which leads to largest overall decrease in loss is natural and interesting. The use case of one group having larger noise and thus the issue of group DRO focusing on that too much is interesting. The idea is a natural solution to that issue.
- The synthetic examples presented in the paper are interesting and clearly bring out the use cases of the method proposed and comparison with group DRO.
- The empirical results presented lead to improved results on a variety of benchmark tasks.

Cons

- I understand the synthetic examples presented but it is still hard for me to understand why this method also works better for the benchmark datasets. For example, in the simple spurious correlation setting with majority and minority groups, I would expect the spurious feature to also have a large reduction in the loss due to the imbalance between majority and minority groups. For group DRO, it makes sense the any large correlation with the spurious feature would hurt one of the groups and thus, the worst loss amongst groups can increase. But, the intuition is not so clear for the method proposed in this paper. Since, the gradient method is not minimizing any fixed loss function, it is hard to understand what is going on. For the real world datasets, it would have been nicer is the paper included some results on why the proposed method did better as it is hard to see why the benchmark datasets reflect the structure proposed in the toy setups.

Questions/Clarifications/Suggestions

- It would be helpful for the readers if the synthetic examples were presented more clearly. For example, I am guessing that the groups are formed randomly in each of the synthetic examples by fixing their sizes. Some figures on how the data actually looks using figures would be helpful for the readers.
- The authors mention their method does not optimize any particular loss function. Is it possible that there exists a loss function which their method optimizes but that is hard to find? What is the precise statement that the authors mean here?
- The authors mention that they have to scale the gradient on page 3 because the norm of the gradient can get very small close to convergence. Can the authors comment more on why scaling by l_i(\theta^t)^p is the right thing to do and how do they choose p=0.5? Why does the scaling need to depend on the loss value at all? Does the analysis in Theorem 1 take into account this scaling or is this an implementation detail?

**Summary Of The Paper:**

The paper gives a new algorithm for the setup where the test distribution is different from the train distribution. The setup includes multiple groups whose information is present during the training time but not during test time and the relative proportion of these groups change during test. The most commonly used method group-DRO does distributionally robust optimization or finds a classifier which performs well on the group with worst loss. This paper proposes to focus instead on the group which leads to maximum decrease in the loss while training instead of the group which has the maximum loss. The paper present several synthetic toy cases where their approach could be useful and concludes with experiments on a variety of benchmarks for this setup and shows improved results.

**Summary Of The Review:**

I would make a recommendation for accepting this paper.

Overall, I think the idea of the paper is interesting and also leads to improved results on benchmark datasets. The main concern is that there could have been more discussion  on why this method works for these real world datasets by connecting them to the synthetic setups presented in the paper.

---

> ### Author Response · Authors · 2021-11-12
> **Response to the weakness/concerns**
>
> Thanks for the feedback, please find our response below.
>
> > For the real world datasets, it would have been nicer is the paper included some results on why the proposed method did better as it is hard to see why the benchmark datasets reflect the structure proposed in the toy setups.
>
> The text benchmarks (MultiNLI, CivilComments-WILDS) and CelebA resemble our toy setup of Sec. 4.3.
> In MultiNLI, the examples with negation words have spurious correlations that negatively transfer between the contradiction and entailment labels. The examples from the group with no negation words do not contain such spurious features.
> Similarly, in CivilComments-WILDS the examples from the black group contain spurious correlation (they contain tokens that identify the demographic, which can be easily exploited to classify examples from black group as mostly toxic), while such spurious features are absent in the non-black group.
> In the CelebA dataset too, male non-blonde (majority) negatively transfers to the male blonde (minority) since the classifier may learn to interpret short hair (male) to be non-blonde. On the other hand, the female blond/non-blond groups do not contain any known spurious correlation.
> In all the cases, CGD could avoid learning spurious features by focussing training on groups with no or relatively low spurious correlation similar to what was demonstrated in Sec 4.3, thereby learning a more robust solution with dampened strength of spurious features.
> FMoW-WILDS, PovertyMap-WILDS are similar to our label noise simple setup of Sec. 4.1.
>
> FMoW-WILDS task is to classify a satellite image into one of 62 land-use categories. The dataset is annotated with human curators labeling if a marked region in an image contains, say a “police station” [1]. Depending on the demographic spread of the human curators, the label correctness is expected to vary from one region to another. Also, some land-use categories are far easier to classify than others (for eg. “police station” vs “helipad”). Similarly, PovertyMap-WILDS task is to map a satellite image to its poverty index. Poverty index per region (urban/rural settlement) ground-truth was acquired through secondary sources such as asset index of the region from the national demographic surveys [2]. The asset to wealth index per region was found to vary per country and hence the quality of the label. The non-uniform label noise of the two datasets is similar to our setup in Sec. 4.1. CGD focuses only on the difficult groups that transfer well to the rest, unlike Group-DRO that only pursue the maximum loss groups.
>
> Thanks for raising this point -- we have added this discussion to the paper.
>
> > Regarding loss function
>
> Sorry for the confusion arising from our writing. Our algorithm CGD indeed optimizes the macro average loss $\mathcal{R}(\theta)$. As Theorem 1 shows, CGD monotonically decreases $\mathcal{R}(\theta)$ and finds its first order stationary points. What we meant was that CGD is not gradient descent (GD), but a different optimization algorithm (just like there are several other optimization algorithms apart from GD such as steepest descent, Newton’s method etc.) for minimizing $\mathcal{R}(\theta)$. We will clarify this in the text.
>
> > Why does the scaling need to depend on the loss value at all? Why is p=0.5?
>
> We wish to normalize the gradients before comparing their relative decrements on the cumulative ERM loss. In addition to normalizing the gradients to be of unit norm, an additional normalization scheme that depends on the loss value works better empirically.
> We scale gradients with their loss raised to p. When we set p to a very large value, the gradient inner products, and hence $\alpha$, of eq. (4) are largely influenced by the loss values. On the other hand, when we set p to 0, we could get stuck in picking low loss train groups repeatedly since we do not update the parameters significantly and hence not converge. In practice, neither of the extremes are ideal. We tried p={0.25, 0.5, 1, 2} on our synthetic setup, WaterBirds, CelebA, and found p=0.5 to perform well empirically.
> We updated Appendix A and some parts of the main paper for better clarity.
>
> > Analysis in Theorem 1 accounts for it?
>
> Theorem 1 accounts for the scaling of the gradients as long as the local quadratic approximation described in Appendix A holds.
>
> > It would be helpful for the readers if the synthetic examples were presented more clearly.
>
> We added figures for synthetic data for better understanding in Appendix C.
>
> **References:**
> 1. Christie, G., Fendley, N., Wilson, J. and Mukherjee, R., 2018. Functional map of the world. In Proceedings of the IEEE Conference on Computer Vision and Pattern Recognition (pp. 6172-6180).
> 2. Yeh, C., Perez, A., Driscoll, A., Azzari, G., Tang, Z., Lobell, D., Ermon, S. and Burke, M., 2020. Using publicly available satellite imagery and deep learning to understand economic well-being in Africa. Nature communications, 11(1), pp.1-11.

---

### Author Response · Authors · 2021-11-12
**Revisions to the paper**

We thank all the reviewers for the valuable feedback. We summarize the changes (changes in the main paper are marked in red) to our paper below.

1. We edited some parts of Introduction and Section 3 for better clarity on our objective and the underlying intuition.
2. We edited parts of Section 2 and Appendix A for more clarity on why the gradient scaling is required, and why we scale them with the square root loss.
3. We added plots that explain the synthetic setting better in Appendix C.
4. We added discussion on how synthetic experiments relate to the real-world datasets in Appendix H.

---

### Author Response · Authors · 2021-11-18
**Gentle reminder of the Phase 1 discussion deadline**

Dear Reviewers,

Could you please take a look at our responses to your concerns and please respond.
We are happy to engage more and clarify any further questions that you may have.

Thanks.

---

### Decision · Program_Chairs · 2022-01-20

**Decision:**

Accept (Poster)

**Comment:**

The manuscript proposes a method for addressing spurious correlations and sub-population (group) shift problem by modelling intergroup interactions. Past work (GroupDRO) focuses on the worst group which is subject to failure when groups have heterogeneous levels of noise and transfer. This work focuses on the group whose gradient leads to largest decrease in average training loss over all groups. The manuscript presents insights on why the proposed method called CGD may perform better than GroupDRO by studying simple synthetic settings. The manuscript also provides empirical evaluation on seven real-world datasets–which include two text and five image tasks with a mix of sub-population and domain shifts.

There are several positive aspects of the manuscript, including:
1. The idea of training on the group which leads to largest overall decrease in loss is natural and interesting;
2. The synthetic examples presented in the manuscript clearly bring out the use cases of the method proposed and comparison with GroupDRO;
3. The empirical results presented lead to improved results on a variety of benchmark tasks.

There are also several major concerns, including:
1. More discussion on why the proposed method works for the chosen real world datasets by connecting them to the synthetic setups presented in the manuscript;
2. The proposed algorithm does not minimize a specific loss function;
3. The standard benchmarks are altered. For example, the CivilComments dataset is shown as a 2 group when it is originally a 8-group task (the groups being the demographics of the users) as shown in the WILDS dataset paper.

Authors clarified, among others, that the proposed approach optimizes the macro-average loss function, and the standard benchmarks are not modified and the experiment setup is exactly like GroupDRO evaluation on the CivilComments-WILDS dataset. Reviewers noted that the generative model has not added anything new since it is essentially the synthetic example and it just shows what every robust machine learning method is supposed to do, i.e., don't rely on e_s (group-specific components) but on e_c (common components) while making predictions. It doesn't justify the procedure of choosing to focus on the group that minimizes the error for the group that decreases all other groups' errors.

The revised manuscript includes a clearer motivation, and more discussion on how the synthetic examples connect to the real world datasets. Based on that, I put an accept recommendation.